# Artistic Recoloring of Image Oversegmentations

Rosa Azami*
Carleton University

David Mould†
Carleton University

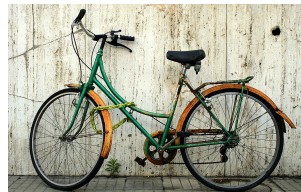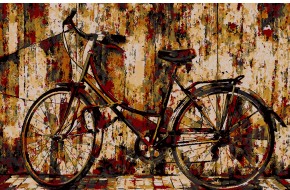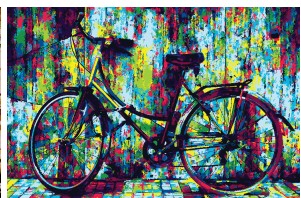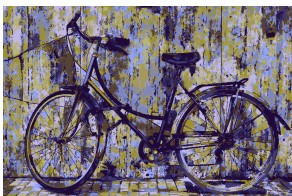

Figure 1: Recoloring of region-based abstraction. The original image is on the left; on the right, we show results from our method using three different palettes.

## ABSTRACT

We propose a method to assign vivid colors to regions of an oversegmented image. We restrict the output colors to those found in an input palette, and seek to preserve the recognizability of structure in the image. Our strategy is to match the color distances between the colors of adjacent regions with the color differences between the assigned palette colors; thus, assigned colors may be very far from the original colors, but both large local differences (edges) and small ones (uniform areas) are maintained. We use the widest path algorithm on a graph-based structure to obtain a spanning tree over the set of regions, then traverse the tree to assign colors in a greedy fashion. Our method produces vivid recolorings of region-based abstraction using arbitrary palettes. We demonstrate a set of stylizations that can be generated by our algorithm.

**Keywords:** Non-photorealistic rendering, Image stylization, Recoloring, Abstraction.

**Index Terms:** I.3.3 [Picture/Image Generation]—; I.4.6 [Segmentation]

## 1 INTRODUCTION

Color plays an important role in image aesthetics. In representational art, artists employ colors that match the perceived colors of objects in the depicted scene. Conversely, abstraction provides the freedom to use arbitrary colors. Figure 2 shows a modern vector illustration of a lion, an example of Fauvism by André Derain, and an abstraction of the Eiffel Tower by Robert Delaunay. The artists have expressed the image content with vivid colors disconnected from the object colors. We aim to generate colorful images, recoloring an image using an arbitrary input palette. Our proposed method uses a subdivision of the image into distinct segments, assigning each segment a palette color in such a way as to preserve visibility of key structures.

Our goal is to present different recoloring possibilities by assigning colors to each region of an oversegmented input image. We aim to maintain the image contrast and preserve strong edges so that the content of the scene remains recognizable. It is important to convey textures and small features, often too delicate to be preserved by existing abstraction methods. We would like to be able to create wild and vivid abstractions through use of unusual palettes.

---

*e-mail: rosa.azami@carleton.ca
†e-mail: mould@scs.carleton.ca

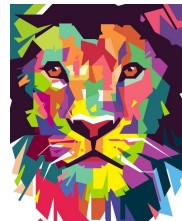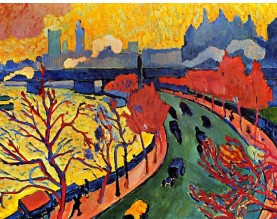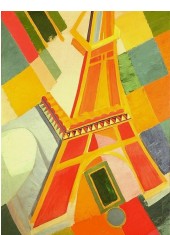

Figure 2: Colorful representational images. A modern vector illustration of a lion; a Fauvist painting by André Derain; an abstraction of the Eiffel Tower by Robert Delaunay.

Manual recoloring of oversegmented images would be tedious. The images contain hundreds or thousands of segments; clicking on every segment would take a long time, even leaving aside the cognitive and interaction overhead of making selections from the palette. We provide an automatic assignment of colors to regions. The assignment can be used as is, could be used in a fast manual assessment loop (for example, if an artist wanted to choose a suitable palette for coloring a scene), or could be a good starting point for a semi-automatic approach where a user made minor modifications to the automated results.

This paper presents an automatic recoloring approach for a region-based abstraction. The input is a desired palette and an oversegmented image. The method assigns a color from the palette to each region; it is based on the *widest path* algorithm [18], which organizes the regions into a tree based on the weight of the edges connecting them. We use color differences between adjacent regions both to order the regions and to select colors, trying to match the difference magnitude between the assigned palette colors and the original region colors. The use of color differences allows structures in the image to remain recognizable despite breaking the link between the original and depicted color.

Our main contributions are the following:

- We designed a recoloring method for an oversegmented image, creating multiple abstractions colored with just one palette. Our method creates wild and high contrast images.

- Various styles can be created by our method. We experiment with color blending between regions and produce smooth images. In addition, we generate new colorings from a palette by applying different metrics and color spaces.

The remainder of the paper is organized as follows. In Section 2, we briefly present related work. We describe our algorithm in Sec-

tion 3. Section 4 shows results and provides some evaluation, and Section 5 gives some possible variations of the method. Finally, we conclude in Section 6 and suggest directions for future work.

## 2 PREVIOUS WORK

Although there is an existing body of work on recoloring photographs [5, 8, 10, 12, 16, 20, 23, 27, 30] and researchers have investigated recoloring in the context of non-photorealistic rendering [2, 7, 19, 24, 26, 31, 32], there is room for further exploration. Existing recoloring methods little address region-based abstractions. The closest approach to ours, by Xu and Kaplan [29], used optimization to assign black or white to each region of an input image. Below, we review some of the previous research on recoloring methods in NPR and color palette selection. These approaches can be broadly classified into example-based recoloring and palette-based recoloring.

### Example based Recoloring (Color Transfer)

Recoloring methods were first proposed by Reinhard et al. [20], where the colors of one image were transferred to another. They converted input RGB signals to Ruderman et al.'s [21] perception-based color space $L\alpha\beta$, then shifted and scaled the $L\alpha\beta$ space using simple statistics.

Neumann and Neumann [16] extracted palette colors from an arbitrary target image and applied 3D histogram matching. They attempt to keep the original hues after style transformation: all colors with the same starting hue should have the same hue after transformation. However, the histogram lacks spatial information and hence is not sufficient for accurate style cloning. They suggest using image segmentation and smoothing the color histogram to improve the result.

Levin et al. [10] introduced an interactive colorization method for greyscale images. They used a quadratic cost function derived from the color differences between a pixel and its neighborhood. User scribbles indicate the desired color in the interior of the region and the colors propagate to the remaining pixels. Inspired by Levin et al. [10], Yatziv and Sapiro [30] presented a method based on luminance-weighted chrominance blending and fast geodesic distance computations. Sykora et al. [26] developed *LazyBrush* for coloring cartoons, integrating textures into images to create 3D-like effects [25]. Casaca et al. [4] used Laplacian coordinates for image division and used a color theme for fast colorization. Fang et al. [7] proposed an interactive optimization method for colorizing hand-drawn grayscale images.

Color compositing has been investigated by NPR researchers. Compositing can be done using alpha blending [14] or the Kubelka-Munk [9] equation (KM), useful in painterly rendering to predict the reflectance of layers of pigment. The data-driven color compositing framework by Lu et al. [12] derived three models based on optimized alpha blending, RBF interpolation, and KM optimization to improve the prediction of compositing colors. Aharoni et al. used a KM-based model for recoloring of styles such as watercolor painting [2], where overlapping stroke layers can produce natural-looking painting effects.

### Palette-based Recoloring

Early methods to extract color palettes used Gaussian mixture models or K-means to cluster the image pixels. Chang et al. [5] introduced a photo-recoloring method by user-modified palettes, using k-means clustering on image colors to obtain palettes. Tan et al. [27] proposed a technique to decompose an image into layers to extract the palette colors. Each layer of decomposition represents a coat of paint of a single color applied with varying opacity throughout the image. To determine a color palette capable of reproducing the image, they analyzed the image in RGB-space geometrically in a simplified convex hull.

Playful Palette [24] is a system for users to create rich palletes, representing a palette as a set of blobs of color that blend together to create gradients. Users manipulate the blobs to obtain the desired range of colors. DiVerdi et al. [6] proposed an approximation of image colors based on the Playful Palette. In this technique, within an optimization framework, an objective function minimizes the distance between the original image and that recolored one by palette colors, based on the self organizing map. The approximation algorithm is an order of magnitude faster than Playful Palette.

Specifically assigning colors to regions has been investigated by previous researchers. Qu et al. [19] proposed a colorization technique for black and white manga using the Gabor wavelet filter; a user scribbles on the drawing to connect the regions, and the algorithm then assigns colors to different hatching patterns, halftoning, and screening. Xu and Kaplan introduced artistic thresholding [29] where an image is segmented, and each segment assigned either black or white color through an optimization process. Lin et al. [11] proposed a palette-based recoloring method with a probabilistic model. They learn and predict the distribution of properties such as saturation, lightness, and contrast for individual regions and neighboring regions, then score pattern colorings using the predicted distributions and color compatibility model of O'Donovan et al. [17]. Bohra and Gandhi [3] proposed an exemplar-based colorization algorithm for grayscale graphic art from a reference image based on color graph and composition matching. They retrieve palettes using the spatial features of the input image. They aim to preserve the artist's intent in the composition of different colors and spatial adjacency between colors in the image.

## 3 RECOLORING ALGORITHM

Our recoloring algorithm automatically assigns colors to regions of an oversegmented image. The system takes as input a set of regions and a color palette and assigns a color from the palette to each region. The recolored image should convey recognizable objects in the image. Edges are essential to the visibility of the structures. Neighboring regions will be assigned distinct colors to express an edge, and regions of similar colors will be assigned similar colors. The human visual system is sensitive to brightness contrast; to help preserve contrast in our recoloring, we take into account the regions' relative luminances when selecting region colors.

We concentrate on color differences between adjacent regions, seeking to match the original color distances without preserving the colors themselves. Neighbouring regions with large differences will be assigned distant colors, preserving the boundary, while similar-colored regions will be assigned similar output colors or even the same color.

We use a graph structure to organize the segmented image, where each region is a node and edges link adjacent regions. To simplify color assignment, we will construct a tree over the graph, with a subsequent tree traversal assigning colors to nodes based on the color of the parent node. Assigning colors along tree paths, rather than using the full graph, increases opportunities for unusual color transitions across unimportant region boundaries, allowing irregular recolorings as in the inspirational artwork of Figure 2. We assign weights to the edges reflecting their priority, with large regions, regions of very similar color, and regions of very different color receiving high priority. Both small regions and regions with intermediate color differences receive lower priority. Once weights have been assigned, we find the tree within the graph that maximizes the weight of the minimum-weight edges, a construction that corresponds to the *widest path* problem.

Our algorithm has two main steps. First, we create a tree by applying the *widest path* algorithm [18] on the region graph. Then we assign colors to regions by traversing the tree beginning from its root. For each region in the graph, we choose the color from the palette that best matches the color difference with its parent.

Before starting, we apply histogram matching between the regions' color differences and the palette color differences. The histogram matching allows us to best convey the image content while using the full extent of an arbitrary input palette, even one that has a color distribution very different from that of the input image.

We show a flowchart of our recoloring approach in Figure 3. The input is (a) an oversegmented image and (b) a palette to use in the recoloring. We compute (c) the adjacency graph of the segments and aggregate the differences between adjacent colors into the set $\{\Delta Q\}$. The widest path algorithm (d) gives us a tree linking all nodes of the adjacency graph. We next compute (e) the set of differences between palette colors, yielding $\{\Delta P\}$. We match the histogram (f) of $\{\Delta Q\}$ to that of $\{\Delta P\}$ and (g) assign colors to all regions by traversing the widest-path tree, resulting in (h) the fully recolored image.

## 3.1  Tree Creation

Before we explain the recoloring proper, we will describe tree creation. We will employ the widest path algorithm to create a tree over the input oversegmentation. Later, we will traverse the tree and assign a color to each region, matching the edge's target color difference with the color difference with its parent's color. In practice, it is possible to combine the tree creation and traversal, since the widest-path algorithm involves a best-first traversal of the tree as it is being built. Prior to color assignment, we apply histogram matching to align the regions' color differences with the palette's for better use of palette colors.

Pollack [18] introduced the widest path problem. Consider a weighted graph consisting of nodes and edges $G = (V,E)$, where an edge $(u,v) \in E$ connects node $u$ to $v$. Let $w(u,v)$ be the weight, called *capacity*, of edge $(u,v) \in E$; capacity represents the maximum flow that can pass from $u$ to $v$ through that edge. The minimum weight among traversed edges defines the capacity of a path. Formally, the capacity $C(u,v)$ of a path between nodes $u$ and $v$ is given by

$$C(u,v) = \min(w(u,a),w(a,b),..,w(d,v)), \tag{1}$$

where $w(u,a),w(a,b),..,w(d,v)$ are the edge weights along the path. The *widest path* between $u$ and $v$ is the path with the maximum capacity among all possible paths.

In a single-source widest path problem, we calculate for each node $t \in V$ a value $B(t)$, the maximum path capacity among all the paths from source $s$ to $t$. The value $B(t)$ is the *width* of the node. The union of widest paths from the source to each node is a tree, which we use to order the color assignment process. We can choose any node as the source; our implementation uses the region containing the image centre.

The widest path algorithm can be implemented as a variant of Dijkstra's algorithm, building a tree outward from the source node $s$ to every node in the graph. All nodes of the graph $t \in V$ are given an initial width value; the source node $s$ will be assigned $B(s) = +\infty$ and all other nodes $v \neq s$ will have $B(v) = -\infty$. A priority queue holds the nodes; at each step of the algorithm, we take the node with the highest current width from the queue and process it, stopping when the queue is empty. Suppose the node $u$ is on top of the queue with a width $B(u)$. For every outgoing edge $(u,v)$, we update the value of the neighbour node $v$ as follows:

$$B(v) \leftarrow \max\{B(v),\min\{B(u),w(u,v)\}\} \tag{2}$$

where $w(u,v)$ is the edge weight between nodes $u$ and $v$. If the value $B(v)$ was changed, node $u$ will be set as its parent node and $v$ will be added to the queue. When the algorithm terminates, all non-root nodes in the graph will have been assigned a parent, thus providing a tree rooted at $s$.

In our application, one possibility for edge weight is to use the difference in color values between the two regions. This would ensure that the widest path tree linked dissimilar regions, resulting

in good edge preservation. However, regions of similar color could easily be divided. We want to preserve small color distances as well, so we need to assign a large edge weight to small color differences. Distances intermediate between large and small are of the least importance. Hence, we base our edge weight on the difference from the median color distance, as follows.

We calculate the color distances across each edge in the adjacency graph; call the set of color distances $\{\Delta Q\}$, with

$$\{\Delta Q\} = \{D_c(c_i,c_j)\} = \{\Delta q_{ij}\}, \quad i \neq j, \quad r_i,r_j \in R \tag{3}$$

where $c_i$ and $c_j$ are the colors of regions $r_i$ and $r_j$, and $D_c$ is the function computing the color distance. Compute the median value $\Delta \bar{q}$ from the distances in $\{\Delta Q\}$.

We also want to take into account the size of the region, such that larger regions have greater importance; we prefer that a larger region have higher priority and thus influence the smaller regions that are processed afterwards, compared to the converse. Depending on the oversegmentation, action may not be necessary; the process is intended to improve results on oversegmentations with a dramatic variation in region size.

We compute for each region a factor $b$, the ratio of the region's size (in pixels) to the average region size. Then, when we traverse an edge, we use the $b$ of the destination region to determine the weight. In our implementation, we compute and store a single edge weight; there is no ambiguity about the factor $b$ because we only ever traverse a given edge in one direction, moving outward from the source node.

To summarize: when traversing an edge, the edge weight is the distance between its target color difference and the median color difference, multiplied by a factor of $(1+b)$ for the destination region:

$$w(r_i,r_j) = (1+b)|D_c(c_i,c_j) - \Delta \bar{q}|. \tag{4}$$

The factor $(1+b)$ takes size into account, but ensures the region's color differences can still affect the traversal order even for very small regions ($b$ near zero). Note that the function $D_c$ depends on the colorspace used. A simple possibility is Euclidean distance in RGB, but more perceptually based color distances are possible. We discuss color distance metrics in section 5.2.

## 3.2  Histogram Matching

We plan to match color differences in the output to the color differences in the input. However, the input palette can have an arbitrary set of colors, and we want to make use of the full palette, which might not happen with direct matching. For example, imagine a low-contrast image recolored with a palette of more varied colors. The smallest palette difference might be quite large; if so, the muted areas of the original will be matched with difference zero, resulting in loss of detail in such regions. A narrow palette applied to a high-contrast image will have similar problems in the opposite direction.

To adapt the palette usage to the input image color distribution, we apply histogram matching to color differences. We emphasize that we are not matching the colors themselves, but the distributions of differences. Histogram matching is applied between the region color differences (the distribution of values in $\{\Delta Q\}$, computed in Section 3.1) and the pairwise color differences of the palette (call this dataset $\{\Delta P\}$).

The histogram matching gives a new target color difference for each graph edge; call this target $\Delta q'(u,v)$ for the edge linking regions $u$ and $v$. The matching ensures that the distribution of values $\{\Delta q'\}$ is the same as the distribution of values in $\{\Delta P\}$. The values $\Delta q'$ are then used for color assignment, selecting color pairs from the palette which correspond to the same place in the distribution: medium palette differences where medium image color differences existed, small differences where the original image color differences were

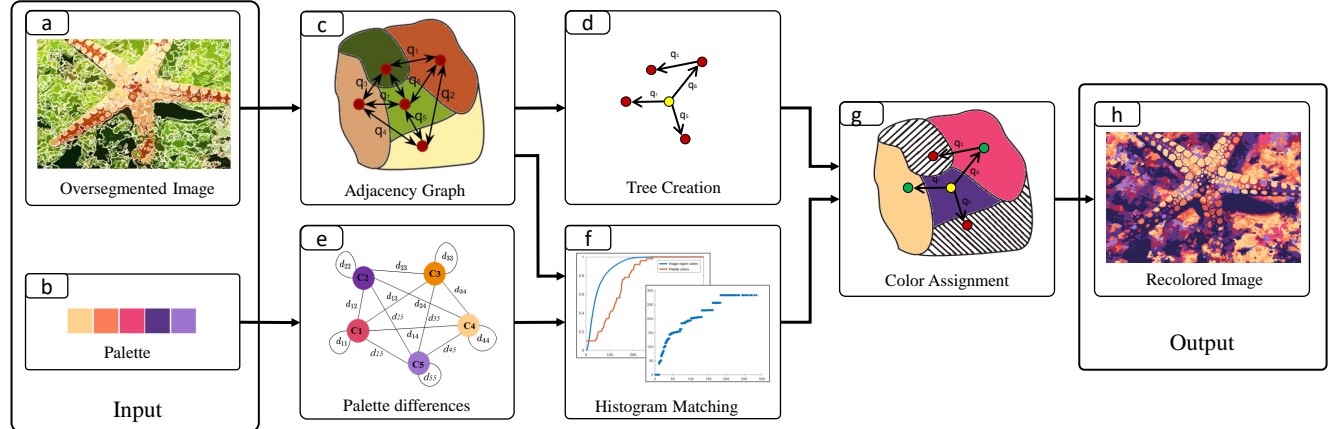

Figure 3: Recoloring algorithm pipeline.

small, with the largest palette differences reserved for the largest differences in the original image. The idea is illustrated in Figure 4, which shows an example of cumulative distribution functions for the original region color differences, the palette color differences, and the region color differences after matching. In this case, the image had smaller differences than the palette, so the target region differences are heightened.

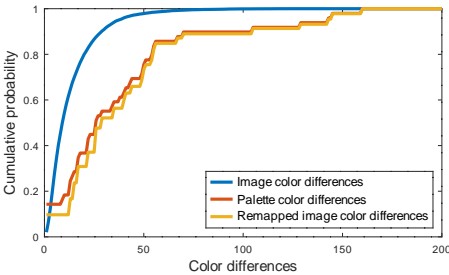

Figure 4: Histogram matching color distances: cdfs of the original image ($\Delta Q$), the palette ($\Delta P$), and the target color differences ($\Delta q'$).

Figure 5 shows the input image with average region colors and the recolored images before and after histogram matching. The bar graphs under each image show the proportion of each color in the image. We can see that the histogram matching used the palette colors more evenly, increasing contrast and highlighting more details. For example, strong edges on the leaf boundary became distinguishable from the nearby regions, and the markings on the lizard became more prominent.

### 3.3 Color Assignment

The widest path algorithm provided a tree, and the histogram matching provided palette-customized target distances for the edges. We now traverse the tree and assign a color to each region along the way. We begin by assigning the closet palette color to the tree's root node; recall that the root was the most central region in the image. At each subsequent step, we assign a color from the palette $P$ to the current region $\alpha$ based on the palette color $p_\beta$ already assigned to the parent region $\beta$ and the target color difference $\Delta q'(\alpha, \beta)$. We also consider the luminance difference between regions $\alpha$ and $\beta$ so as to help maintain larger-scale intensity gradients. Recall our intention in preserving color differences: two regions with large color differences should be assigned two very different colors, and regions with a small color difference should get very similar colors, possibly

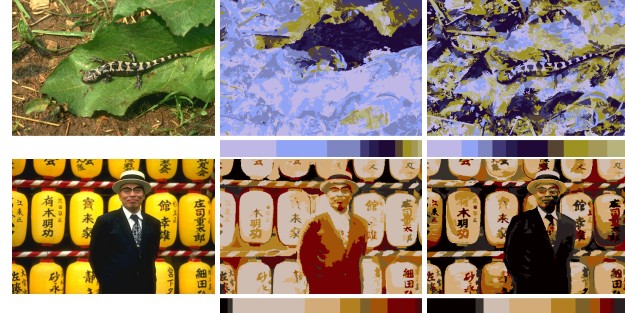

Figure 5: Histogram matching result. Left to right: original image, result without histogram matching, and result using histogram matching.

the same color. Owing to histogram matching, "large" and "small" are calibrated to the content of the particular image and palette being combined.

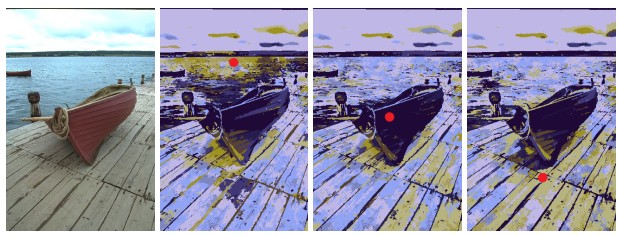

Figure 6: Changing the source region locations. The starting region is indicated by a dot.

We impose a luminance constraint on potential palette colors, in an effort to respect the relative ordering of the regions' luminances. Suppose the luminances of two regions $\alpha$ and $\beta$ are $L_\alpha$ and $L_\beta$, where $L_\alpha < L_\beta$. We then constrain the set of eligible palette colors for region $\alpha$ such that only colors $p_\alpha$ that satisfy $L_{p_\alpha} < L_{p_\beta}$ are considered. A similar constraint is imposed if $L_\alpha > L_\beta$.

For region $\alpha$ and its parent $\beta$, we have the target edge difference $\Delta q'(\alpha, \beta)$. Denote by $p_\beta$ the palette color already assigned to the parent region $\beta$. We choose the palette color $p_\alpha$ for region $\alpha$ so as to minimize the distance $D$:

$$D = |\Delta q'(\alpha, \beta) - D_c(p_\alpha, p_\beta)| \qquad (5)$$

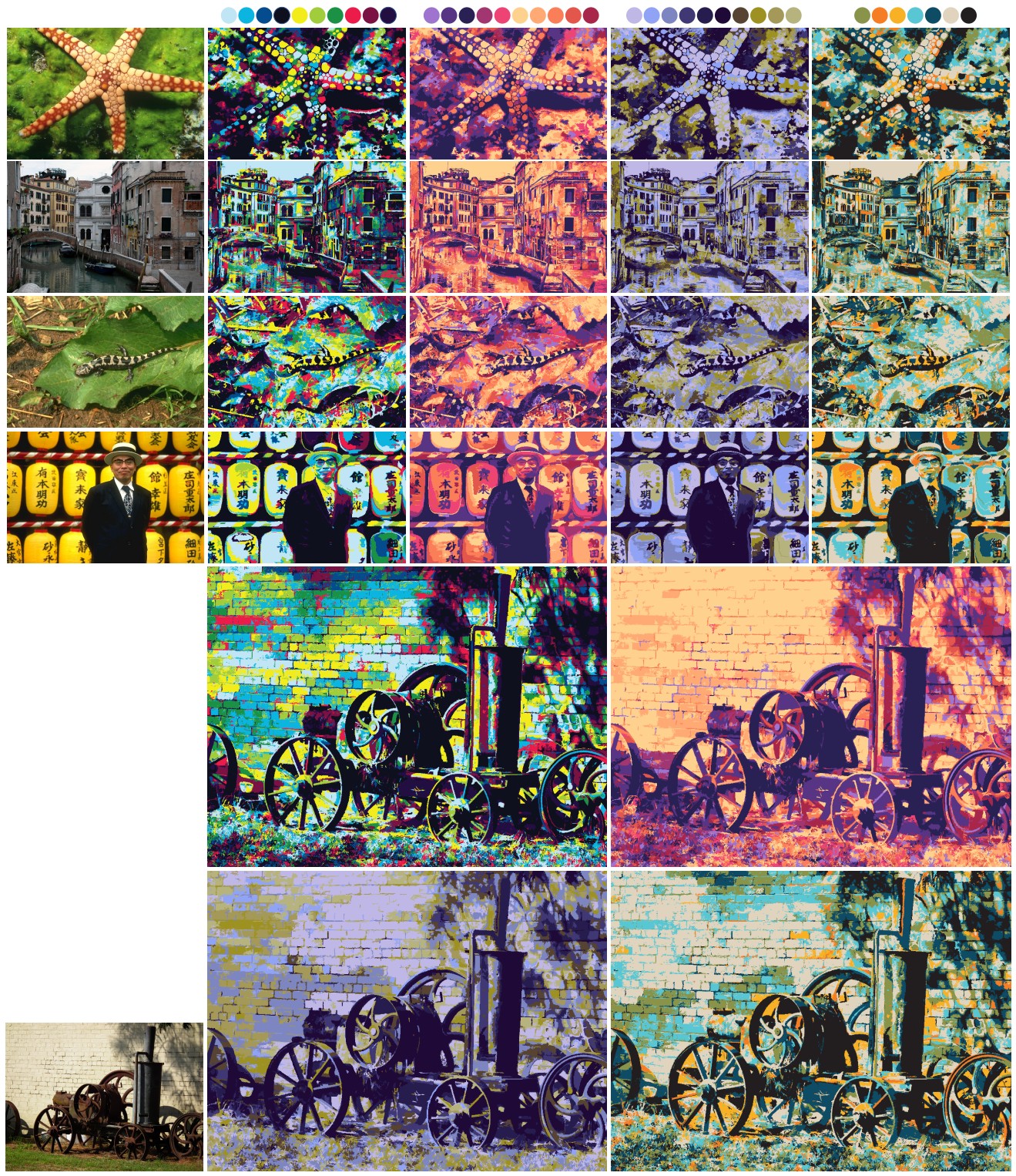

Figure 7: Region recoloring results. Top row: visualization of palettes used. Images, top to bottom: *starfish*, *Venice*, *lizard*, *lanterns*, and *rust*.

where $D_c$ is the distance metric between two colors.

For colors with the lowest and highest luminance in the palette, there may be no available colors satisfying the luminance constraint. In such cases, the constraint is ignored and all palette colors are considered.

Since the source region has no parent, the above process can not be used to find its color. Instead, we assign the closest color from the palette, as determined by the difference metric $D_c$. The source region itself is the region containing the centre of the image; while the output is weakly dependent on the choice of starting region, we do not view the starting region as a critical decision. Figure 6 shows some examples of varied outcomes from moving the starting region.

## 4 RESULTS AND DISCUSSION

Figure 7 shows a variety of recolored images generated by our algorithm using various palettes. We succeeded in maintaining strong edges, and objects in the recolored abstractions remain recognizable. Our algorithm retains textures and produces vivid recolored images by selecting varied colors from the palette. It assigns the same colors over flat regions and distinct colors to illustrate structures. We ran our algorithm on a variety of images with different textures and contrasts. We obtained most of our palettes from the website *COLRD* (http://colrd.com/); others we created manually by sampling from colorful images.

In Figure 7, we present a set of examples from our recoloring algorithm, which were generated with four different palettes. We chose images presenting different features. The delicate features and textures in the abstractions stay visible after recoloring despite the input photographs having been radically altered by the recoloring.

In the *starfish* image, the structure and the patterns on the arms become more prominent. The uniform colors of the background become a vivid splash of colors, emphasizing the textureness of the terrain. The algorithm has chosen the darkest colors to assign to the shadows and the lightest ones to the surface of the creature.

The *Venice* canal is a crowded image composed of soft textures and structures with hard edges. The algorithm is able to preserve recognizable objects such as the boats and windows. Even tiny letters on the wall and pedestrians on the canal's side are visible. The recoloring process preserved the buildings' rigid structures; meanwhile, it captured shadows and the water's soft movements. In presenting such features, adopting a highly irregular oversegmentation was necessary.

The *lizard* image is an example of a low-contrast image with textured areas covered by dull colors. The algorithm highlighted the textures by assigning wild colors to the homogeneous regions on the leaf. At the same time, substantial edges like the lizard's body patterns and the leaf edges are preserved naturally by our algorithm.

The next example shows a high-contrast input. The algorithm assigned the darkest colors from each palette to the coat of the man and separated it from the background using a very light color. Further, the small features on the face and the Chinese characters are mostly readable.

The *rust* image shows different textures on the wall and the grass, plus soft textureless areas on the machinery. The brick patterns on the wall, exaggerated in the color assignment, made the final images more interesting than the original flat image. The high-frequency details of the grass are retained. The smooth transition of colors on the top right of the image portrays the shadows.

We demonstrated strong edge preservation in all examples. Additionally, the image textures were preserved, and palette colors are uniformly used to maintain a good contrast.

### 4.1 Comparison with Naïve Methods

Figure 8 gives a comparison between our method and two naïve alternatives: closest color and random color. For "closest color", each segment is colored with the palette color closest to the segment's average color. For "random color", each segment is randomly and independently assigned a color from the palette. Using the closest color preserves some image content, but the result shows large regions of constant color; many of the palette colors are underused, an issue that can worsen when there is a significant mismatch between the original image color distribution and the palette, as in the upper example. Random assignment provides an even distribution of palette colors, but the image content can become unrecognizable for highly textured images, as in the lower example. Our method uses the palette more effectively, showing local details and large-scale content and exercising the full range of available colors.

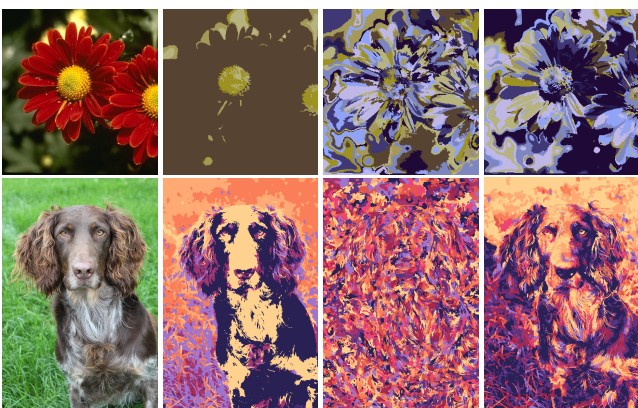

Figure 8: A comparison with naïve recoloring. Left to right: The original image, the results from closest color, random color, and the proposed method.

### 4.2 Comparison with ColorArt

We next compare our recoloring method with *ColorArt* [3], an optimization-based recoloring method for graphic art. This method assigns colors to regions by solving a graph matching problem over color groups in the reference and the template image. In searching for a reference image, this algorithm uses the same number of color groups as in the template image.

Figure 9 shows images generated by *ColorArt* on the right and ours in the middle, both using the *sunset* palette. We created a colorful leaf surrounded by a light background as in the input image, showing the algorithm respects the changes in lightness. Moreover, assignment of different colors on the leaf presented an interesting texture. The leaf image generated by the *ColorArt* method has reversed the image tones. In the *sketch* image, we preserved the edges and showed a recognizable face in the image. In contrast, the *ColorArt* algorithm had difficulty with the edges and the gradual gradients, resulting in a somewhat incoherent output.

### 4.3 Recoloring with SLIC0 Oversegmentation

Our recoloring algorithm does not make any assumptions about the input oversegmentation. Figure 10 shows results from an oversegmantation from SLIC0 [1]. The *starfish* and *owl* images have approximately 2000 and 5000 segments, respectively.

Note that more irregular regions can better represent complex image contours and textures, allowing the recolored abstractions to better display the image content. In *starfish*, the structures and shadows are represented by distinct colors that contrast between the object and the background. The strong edges, such as the arms of the starfish, are preserved; however, the thin features are not captured by SLIC0's uniform regions, and the background terrain does not present any significant information. In the *owl* image, the small regions on the chest convey the feather textures, while regions such as dark eyes kept their well-defined structures. Given a suitably detailed oversegmentation, we can produce appealing results.

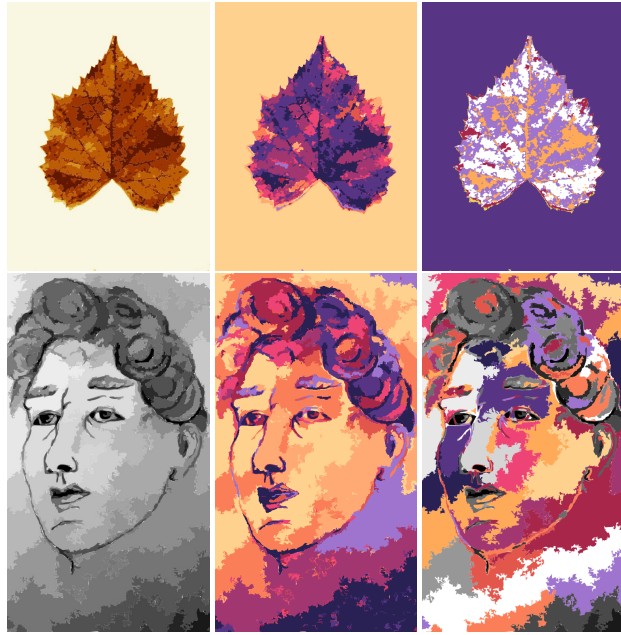

Figure 9: Comparison with ColorArt. Left: input; middle: our results; right: ColorArt results.

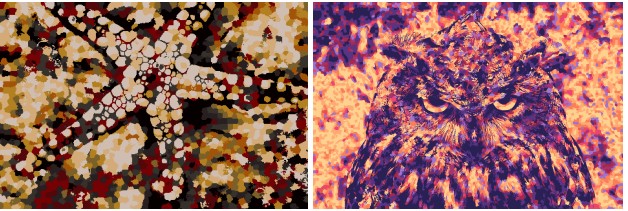

Figure 10: Recoloring with SLIC0 oversegmentation.

## 4.4 Performance

We ran our algorithm on an Intel(R) Core(TM) i7-6700 with a 3.4 GHz CPU and 16.0 GB of RAM. The processing time increases with the number of regions and edges in the graph. The time complexity of single-source widest path is $\mathcal{O}(m + n \log n)$ for $m$ edges and $n$ vertices, using a heap-based priority queue in a Dijkstra search.

Table 1 shows the timing for creating the trees and color assignments of different images. For small images like the *starfish*, containing about 1.4K regions, the recoloring algorithm takes about 0.007 seconds to construct the tree, while it takes about 0.3s for larger images such as *rust* with 7.2K regions. With a palette of 10 colors, the color assignments take 0.05s and 1.2s to recolor the *starfish* and *rust* respectively. The color assignment will take longer for larger palettes and images with a larger number of regions. We show the timing of tree creation and color assignments for all images in the gallery.

Table 1: Timing results for images with varying numbers of regions.

| Image | Tree creation | Color assignment | Graph | Total | # Regions |
|---|---|---|---|---|---|
| lanterns | 0.003s | 0.02s | 0.064s | 0.087s | 1K |
| starfish | 0.007s | 0.05s | 0.07s | 0.127s | 1.4K |
| lizard | 0.015s | 0.08s | 0.1s | 0.195s | 1.9K |
| rust | 0.3s | 1.2s | 0.9s | 2.4s | 7.2K |
| Venice | 0.3s | 1.4s | 0.9s | 2.6s | 7.7K |

## 4.5 Limitations

Although in our experience our method works well for most combinations of image and palette, there are cases where the output is unappealing. When two similar regions are not neighbours, they may receive different colors; e.g., a sky area may be broken up by branches and different parts of the sky could be colored differently. Even adjacent regions may not receive similar colors if their average colors differ, introducing spurious edges into regions with slowly changing colors such as gradients or smooth surfaces. Out-of-focus backgrounds and faces are common examples producing such effects,

Figure 11 shows two failure examples. The woman's face is given an irregular, high-contrast color assignment, and her eyes look sunken. The busy background has similar contrast levels to the face, making the overall composition unappealing. In the Étretat results, large regions of different colors appear in the sky, which does not look attractive. Because the original regions have different average colors, our algorithm is likely to separate them regardless of the palette.

Our algorithm is at present strictly automated with no provision for direct user control beyond choice of palette and parameter settings. While these parameters provide considerable scope for generating variant recolorings, so that a user would have a wide range of results to choose from, direct control is not yet implemented. One might imagine annotating the image to enforce specific color selections or linking regions to ensure that their output colors are always the same. While it would be straightforward to add some control of this type, we have not yet implemented such features.

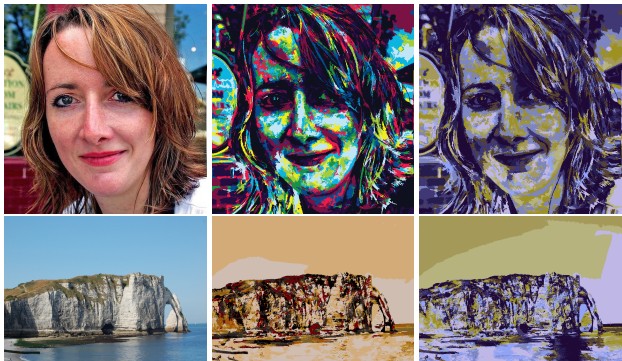

Figure 11: Failure cases. Above: woman; below: Étretat.

## 5 VARIATIONS

Previously, we strictly adhered to the palette colors. Another possibility is to obtain intermediate colors; here, we apply spatial blending to produce smooth color transitions. In the following subsection, we show how different color spaces and distance metrics can generate various results from the same palette.

### 5.1 Blending

In transferring the colors, we have intended to strictly preserve the palette and not add colors. However, we can also blend the colors, giving a more painterly style. Blending introduces new intermediate shades of colors. We suggest cross-filtering the recolored image with an edge-preserving filter such as CRGF [15]. This process smooths the areas away from edges while maintaining strong edges.

The cross-filtering mask size will affect the outcome. Larger masks will produce a stronger blending effect; small features will be smoothed out, and the output image will become blurry in regions lacking edges. Figure 12 illustrates examples of blending using masks of sizes $n = 20$, 100, and 300. Blending with $n = 20$ only

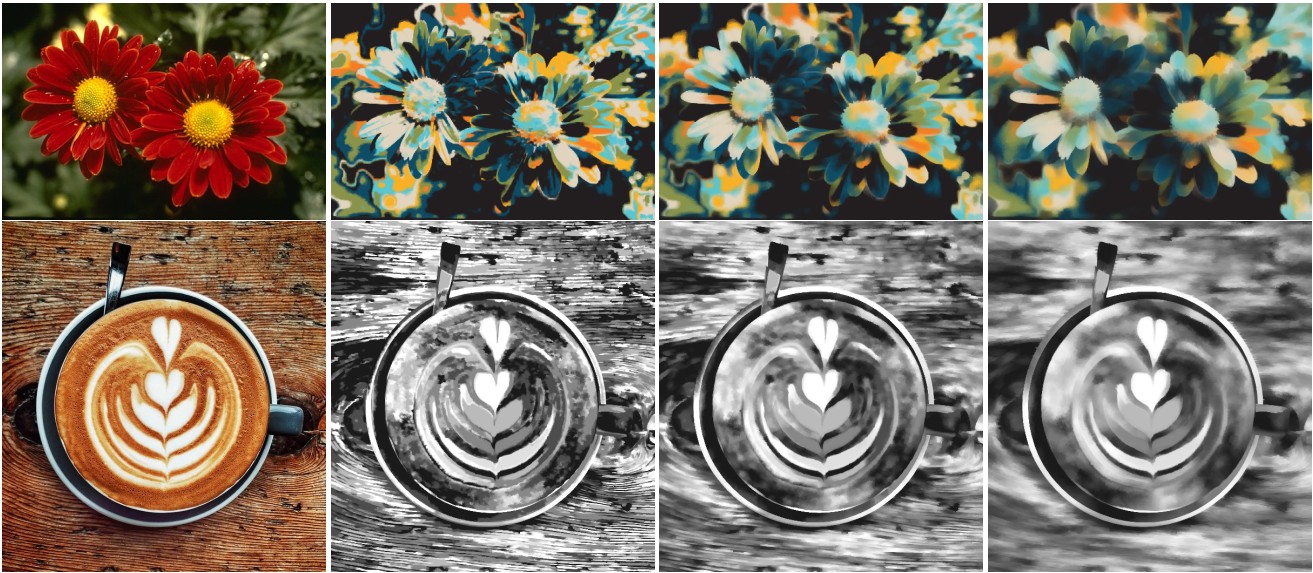

Figure 12: Postprocessing with an edge-aware filter. Left to right: original image; cross-filtered images with mask size 20, 100, and 300.

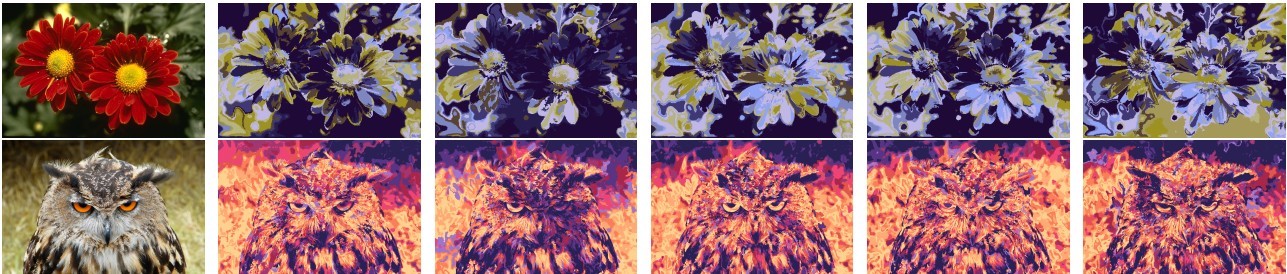

Figure 13: Recoloring with different distance measures. Left to right: Input image, results from RGB, CMC, CIE94, CIE2000, and CIE76 color distances.

slightly modifies the image; for larger masks, the blending is more apparent. At $n = 300$ we can see a definite blurring in originally smooth areas, although blurring does not happen across original edges. Using a gray palette, can can obtain an effect resembling a charcoal drawing with larger masks.

### 5.2 Color Spaces and Distance Metrics

We can employ different functions for our color distance function $D_c$. Different choices of color space and distance metric can affect the recoloring results. Changing the distance metric will cause both the widest-path tree and the color assignment to change.

We have experimented with computing color distances with the Euclidean distance in RGB as well as using perceptually uniform measures CIE94, CIEDE2000, CIE76, and CMC colorimetric distances [22, 28].

Both CIE94 and CIEDE2000 are defined in the Lch color space. However, CIE94 differences in lightness, chroma, and hue are calculated from Lab coordinates. CMC is quasimetric, designed based on the Lch color model. The CIE76 metric uses Euclidean distance in Lab space.

In Figure 13, we show different outcomes from different metrics using two palettes. We can observe the strong edge and contrast preservation, which is an apparent result of perceptual uniform metrics. More importantly, each metric gives a different variant, which allows a user to choose from different results. We can get interesting results from each metric. However, in our judgement, more attractive results are obtained from RGB and CMC colorspaces; the delicate features and image contrast are maintained, and objects are generally preserved. CIE94 and CIE2000 metrics are also effective, but we found that the CIE76 metric rarely creates interesting results.

## 6 CONCLUSIONS AND FUTURE WORK

In this paper, we presented a graph-based recoloring method that takes an oversegmented image and a palette as input, and then assigns colors to each region. The result uses the palette colors to portray the image content, but without attempting to match the input colors. Designing our algorithm with the widest path allowed us to maintain the image contrast and objects' recognizability. We demonstrated our results with different palettes. We achieved vivid recoloring effects, effective for most combinations of input images and palettes.

In the future, we would like to investigate non-convex palette color augmentation, adding new colors extending an input palette while matching the palette's theme. We would like to extend the color assignment to consider color harmony, scoring based on compatibility of colors and thus effecting the ability of certain colors to be neighbors. Furthermore, we would like to be able to recolor smoothly changing regions like the sky more uniformly. Adding elements of user control would allow for better cooperation between the present automated method and the user's intent.

### ACKNOWLEDGMENTS

Thanks go to members of the GIGL group and to the anonymous reviewers for helpful suggestions. The authors thank NSERC and

Carleton university for financial support.

Many images were provided by photographers who made their work available through Flickr. Thanks to the following contributors: Esteban Chiner (bicycle), Mark & Lesley (owl), Pug Girl (Venice), Carl Roi (woman), sv1ambo (rust), Michal Ščuglík (dog), Taymaz Valley (coffee), and daameriva (Étretat). The "sketch" and "leaf" images were taken from freesvg.org; the "lion" image is from Vecteezy.com user danvectorman. Other images (lanterns, starfish, lizard, flower, boat) came from The Berkeley Segmentation Dataset and Benchmark [13].

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
