# OpenReview forum: "Artistic Recoloring of Image Oversegmentations"
_graphicsinterface.org/Graphics_Interface/2021/Conference/Second_Cycle — GI 2021_

### Official Review · Reviewer_nLZB · 2021-05-03
**Artistic Recoloring, paper 27**

**Rating:** 6
**Confidence:** 2

**Review:**

This paper proposes a novel technique for artistic recoloring of an image. The pipeline consists of over-segmentation and solving the "widest path problem" for creating a tree for color assignment. In addition, the input palette is extracted and palette elements are assigned via histogram matching. This paper demonstrates results on various images, compares to ColorArt and naive baseline of assigning the closest color to each segment, as well as shows various ablations. On the pro side, the method is simple, easy to implement, and results look interesting. On the con side, I think the contribution is really marginal, and it's not hard to think of stronger baselines (e.g., solve the problem as graph-cut, as in Lau et al. 2011)

[Lau et al. 2011] 	C. Lau, R. Mantiuk, W. Heidrich. Cluster-based Color Space Optimizations. ICCV, 2011

---

### Official Review · Reviewer_6ZEv · 2021-05-04

**Rating:** 5
**Confidence:** 3

**Review:**

This paper proposes an approach to recoloring oversegmented images given an input palette. It has a multi-stage approach that includes one-to-many widest-path computation, a best-first tree traversal, histogram matching, and constraint-based color assignment. It demonstrates recoloring on a variety of qualitative examples and quantitatively measures runtime.

Overall, it is an interesting approach that can produce compelling images. However, it is missing convincing experimental support. First, there is important related work that is not adequately handled. Second, the pipeline is unusual and it lacks the ablations necessary to justify it. The recommendation is 5 (marginally below) but depending on the weight given to the interesting and seemingly unique qualitative results it could be higher.

+The approach can produce some subjectively very nice looking stylized images. Especially in the supplemental material, there are some compelling images (the skyscraper hallway, the cat, the car, etc.).

+The blending effect in section 5.1 produces some nice results. Particularly the top right image is very nice.

+The paper is fairly unique in its approach (oversegmentation, widest path selection, etc.) compared to other known related work. Regardless of the other downsides of this discussed below, like optimality, there are real benefits to an approach that is unique in this context. Especially for such a subjective task, having a breadth of different approaches in the literature is very valuable. Practically it is valuable because more available filters in an image editor are a good thing if they can produce a distinct visual effect. It is also useful because such novelty is essential to creating new research directions.

-Recoloring and color palette transfer are well-studied tasks, but this paper compares to only one other method (ColorArt, which is designed for graphic art) using only two qualitative comparisons. Other papers, like Chang et al. 2015 [5] and Tan et al. [26] seem to be critical related work that are not addressed. They are cited and summarized in the related work, but with no justification for why they do not solve the problem and no experimental comparison. How do these methods compare to this approach? It seems they can recolor robustly based on palettes and don't require an oversegmentation (but an oversegmentation could be generated afterwards if desired).

-This paper focuses on recoloring an oversegmented image. Why is the oversegmentation a part of the task, rather than the method? This is an important distinction for the related work, but no clear motivation for focusing exclusively on oversegmented images is given. This method does assign colors to all pixels, the choice of superpixel size seems like an important parameter, and existing full-image methods can seemingly do oversegmentation palette transfer by doing full image transfer and then oversegmenting (or the converse, if desired).

-The core algorithm picks an arbitrary source superpixel, runs widest path to all sinks, generates the tree defined by their union, and then uses a best-first traversal of that tree for a greedy coloring of the image. This is roundabout and avoids clearly defining the task objective and optimizing it. The problem as framed is an overconstrained graph coloring problem with color-difference-based weights. Why use a multi-stage heuristic instead of directly solving an optimization with standard techniques? For example, simulated annealing or integer programming (and corresponding LP relaxations) could directly optimize the graph coloring quality. There are also simpler greedy heuristics, like bottom-up clustering. This isn't to say that the widest path approach proposed here is inferior. The concern is that widest path is core to the approach but isn't well justified with ablations or strong baselines. Why is this the right solution?

-The "naive" baseline comparisons are too naive. A random assignment of colors to segments is highly suboptimal. The paper would be stronger if there were more comparisons to existing methods and more comprehensive ablations instead of these weak baseline comparisons.

-The paper requires fairly extensive copy editing.

---

### Official Review · Reviewer_4Bat · 2021-05-04
**Pretty results, decent method, well written**

**Rating:** 7
**Confidence:** 3

**Review:**

The paper proposes a new method to automatically recolor an oversegmented image/photograph given a palette. Treating regions as graph nodes and adding edges between adjacent ones, they add an edge cost composed of color difference and size of one of the regions. Then they fix a single region as a root/source, find distances to each graph vertex via widest path (which is just l_\infty shortest path), forming a tree. They then adjust the histograms to make the distributions of color in the image and the palette similar, and finally assign the colors per each region greedily traversing the tree, using the previously found color differences. They validate the method via visual inspection and comparison to naïve methods and ColorArt.

I found the paper very well written and easy to follow. The method, while not terribly novel or well justified, makes sense overall, is fast, and produces beautiful results. I really enjoyed looking at the final colorizations. Therefore, I vote for a rather clear 'accept', and I have just a handful of comments that I would like to see addressed.

* I am unclear how histogram matching works. I didn't see the actual algorithm in 3.2, and the only visualization of the process I found was in the overview Fig.3. Maybe it's worth adding concrete steps and a figure? Or referring to an existing work that does something similar?
* I am not sure why the authors preferred to use greedy color assignment following the tree in 3.3, rather than an optimization. Optimization could actually use the tree the authors computed, defining a 'derivative' (D) operator, and we'd need to minimize something like equation 4, but globally for the whole image. It probably wouldn't make much visual difference, I don't know, but might work better than the greedy assignment?
* The widest path logic is not super clear. Why should each path have minimum l_inf norm, and not the tree be, say, min spanning tree?

Minor things:
* At the beginning of Sec.3: "cotaining"
* The font for maths has a different size that the rest of the text.

---

### Meta-Review · Area_Chair_PRPg · 2021-05-04

**Recommendation:** Accept
**Confidence:** 4

**Metareview:**

All reviewers agree that the method has some merit and novelty. It is well-described, easy to reproduce, and produces interesting and novel results. Reviewers are also consistent on weaknesses of the paper:
 - some technical choices are sub-optimal (e.g., greedy color assignment), or lack clear motivation (e.g., using widest path, or over-segmented inputs), and
 - baselines are weak.

Given that chairs explicitly ask to favor novelty over rigorous evaluation, I think this paper is suitable for GI 2021.

---

### Decision · Program_Chairs · 2021-05-08

Accept